# Design, synthesis, and *in vitro* evaluation of a carbamazepine derivative with antitumor potential in a model of Acute Lymphoblastic Leukemia

**Cristian Álvarez-Gómez, Angela V. Fonseca-Benítez, James Guevara-Pulido** *

INQA, Química Farmacéutica, Universidad El Bosque, Bogotá, Colombia

* joguevara@unbosque.edu.co

**Citation:** Álvarez-Gómez C, Fonseca-Benítez AV, Guevara Pulido J (2025) Design, synthesis, and *in vitro* evaluation of a carbamazepine derivative with antitumor potential in a model of Acute Lymphoblastic Leukemia. PLoS One 20(4): e0319415. https://doi.org/10.1371/journal.pone.0319415

## Abstract

Acute lymphoblastic leukemia (ALL) is a significant concern in both pediatric and adult demographics. Despite 156 approved cancer therapies based on small molecules, a mere five apply to all types of leukemia. Unfortunately, adherence to these treatments is low due to adverse side effects. Consequently, there is an urgent need to identify more effective treatment options for ALL. This study presents a potential solution. We have designed over fifty analogs of carbamazepine, utilizing a combination of ligand-based and structure-based drug design methodologies. Among these analogs, we identified the CR80 analog, which demonstrated predicted binding values of -8.66 kcal/mol against beta-tubulin, a favorable LogP, and $IC_{50}$ values suitable for *in vitro* evaluation. The CR80 compound was synthesized with a yield of 50% and subsequently assessed *in vitro* against the U-937 cell line. It obtained an $IC_{50}$ value of 0.8 micromolar to 1 micromolar and a selectivity index of two, thus marking it as a promising candidate for *in vivo* studies.

## Introduction

Acute lymphoblastic leukemia (ALL) is the most common pediatric cancer. It is the second most common acute leukemia in adults, with an incidence of over 6500 cases per year in the United States alone [1]. The American Cancer Society estimates that in 2023, the United States will see over 6,500 new cases and almost 1,400 deaths from acute lymphoblastic leukemia (ALL). Sixty percent of cases occur in children, peaking at ages 2–5 and another after age 50 [2].

While the number of approved cancer treatments based on small molecules is 156, only five are for all leukemia types[3]. Among these, 3 are biological drugs, and 5 are small molecules. The prognosis for patients with acute lymphoblastic leukemia (ALL) has dramatically improved due to intensive multimodal treatment strategies, such as chemotherapy, high-dose chemotherapy with stem cell rescue, and

**Data availability statement:** All relevant data are within the paper and its Supporting Information files.

**Funding:** The author(s) received no specific funding for this work.

**Competing interests:** The authors have declared that no competing interests exist.

radiation therapy when necessary [4]. The treatment of Adult Acute Lymphoblastic Leukemia (ALL) involves complex chemotherapy combinations and schedules typically seen in oncology. Two main chemotherapy regimens are currently used. The Berlin-Frankfurt-Münster protocol features an induction regimen, consolidation regimen, intensification regimen, and maintenance therapy, primarily implemented in European adult ALL clinical trials. Alternatively, the hyper-CVAD regimen, created by MD Anderson Cancer Center researchers, consists of rotating two intensive chemotherapy cycles [5]. However, some treatments do not reach the market due to their toxicity [6]. The current landscape of cancer treatments is marked by their unpleasant and highly toxic side effects [7]. For ALL treatments, patients may experience disorders related to the blood and lymphatic system, eyes, gastrointestinal system, liver, metabolism, musculoskeletal system, respiratory system, and skin, as well as injuries, poisonings, and complications of procedures. This can lead to symptoms such as headaches, weight fluctuations, diarrhea, nausea, and vomiting due to the lack of specificity of these treatments for cancer cells, which can lead to poor treatment adherence [8–11]. The urgency of finding better ALL treatments is apparent. Our study presents a potential solution. The design, synthesis, and *in vitro* evaluation of potential antitumoral carbamazepine derivatives offer a promising path to a more effective and less toxic ALL treatment.

Recent research aims to develop drugs offering targeted treatments with fewer side effects and improved patient adherence. Computational drug discovery strategies, such as Computer-Aided Drug Design (CADD), are now used to identify, design, and optimize compounds through artificial intelligence and molecular modeling. This multidimensional data integration allows for improved use of time and resources, ultimately enhancing the development of new drugs [12–14]. CADD is a powerful tool that accelerates the repositioning of drugs by identifying potential new uses for existing molecules that have passed safety and toxicity tests and are already on the market. Carbamazepine (CBZ), an anticonvulsant and neuropathic pain medication used for over 20 years, has shown effects on the replication of hematopoietic cells since 1995 [15], leading to a 50% decrease in blood cell count. In more recent studies, Meng et al. (2010) demonstrated that carbamazepine promotes the degradation of the Her-2 protein in breast cancer cells by modulating HDAC6 activity and acetylation of Hsp90 [16].

Additionally, in 2020, Zhao reported that CBZ has an affinity for the Frizzled FZD8 receptor (via Wnt); inhibiting this receptor decreases bone remodeling and promotes the apoptosis of bone cells, which are the origin of some types of leukemia [17]. Based on the information provided, the pharmacophoric core of CBZ has shown intriguing potential for treating ALL. This research utilized CADD strategies, such as structure-based drug design (SBDD) and ligand-based drug design (LBDD), to improve the pharmacodynamics of newly designed carbamazepine analogs. In the LBDD approach, QSAR models were constructed using the INQA-Artificial Neural Network, which our group had previously validated in developing new SSRIs, AKR1C3, and JAK-3 drugs [18–20]. This allowed us to predict the IC$_{50}$ values and forecast the pharmacokinetic values of the newly designed candidates.

Subsequently, these candidates were synthesized and evaluated *in vitro*, transitioning from *in silico* design to *in vitro* testing.

## Materials and methods

### Computer-aided drug design

To design molecules rationally, we conducted a literature search in public databases such as ChEMBL, DrugBank, and PubChem to find molecules that have shown biological activity against ALL and beta-tubulin. Then, we used two strategies for the chosen molecules.

### Structure-based drug design

The binding affinities of drugs that affect beta-tubulin, including their designed analogs and other frequently prescribed medications for treating ALL, were evaluated in kcal/mol utilizing the PDB 6QUS crystal structure [21]. Protein preparation followed the AutoDockTools protocol [22]. The co-crystallized paclitaxel ligand was removed using Samson software [23]. After preparing the crystal, docking was conducted with the known active ligand, vincristine. Following validation of the docking, additional energies were calculated. The structures were mod[eled, and their energies were optimized in Avogadro [24] using the MMFF94s force field. Subsequently, 35 drugs and 58 designed analogs were docked with 6QUS in AutoDock Vina. The grid box was set at 13 × 15 × 25 points with a grid spacing of 0.375 Å, centered at coordinates 2, 23, and 2. Calculations were conducted in triplicate, and the affinity energy of the pose with the lowest RMSD value was averaged for each compound. The interactions and distances were visualized using Discovery Studio Suite®.

### Ligand-based drug design

We used the INQA-Artificial Neural Network (INQA-ANN) architecture to create a predictive QSAR model [25]. This model correlates molecular descriptors with experimental $IC_{50}$ values for beta-tubulin and ALL drugs found in the literature. The goal was to predict the $IC_{50}$ value of 58 designed analogs. Initially, we calculated molecular descriptors using PaDEL-Descriptor v2.20 software [26]. The obtained descriptors were then evaluated through Pearson correlation. As a first step, the descriptors were grouped into subfamilies, and any descriptor with a correlation value between 0.2 and -0.2 was selected. Subsequently, a new Pearson correlation was performed with the filtered descriptors and the experimental $IC_{50}$ values. The descriptors with correlation values closest to 1 were ultimately chosen. Then, six molecular descriptors were selected as inputs, and 21 $IC_{50}$ values of the molecules reported in the literature were used as outputs. The number of nodes in the hidden layer was gradually adjusted during predictions, starting with around 100. We selected the model with a coefficient of determination ($R^2$) exceeding 0.7, as calculated by INQA-ANN [27]. Additionally, this model was statistically validated using QSAR validation parameters cited in the literature, including k, R, and the correlation of these values [28].

### Chemistry

All reagents used in the experiment were obtained from commercial suppliers and used without further purification. To monitor the progress of the reaction, TLC was performed on aluminum plates coated with silica gel F254 indicator, which was visualized by UV irradiation. Flash chromatography used silica gel 60 (230–240 mesh). For NMR analysis, [1]H and [13]C spectra were recorded in $CDCl_3$ using a Bruker Avance NEO 400 MHz spectrometer. The chemical shifts for [1]H and [13]C were indicated in parts per million (ppm, δ), with tetramethylsilane as the internal reference. The splitting patterns for [1]H NMR were designated as singlet (s), doublet (d), triplet (t), quartet (q), and multiplet (m). The coupling constants and integration were quoted in Hertz (J). Infrared spectra were recorded using a Bruker Alpha-P ATR FTIR with a diamond crystal. High-resolution mass spectrometry was carried out using an Agilent 5973 (80 eV) spectrometer with electrospray ionization (ESI).

The synthesis was carried out using a round-bottom flask connected to a reflux system. 0.5 mmol (118 mg, 99%) of carbamazepine was added and reacted in 5 mL of THF for 3 hours at reflux. A catalyst of 10 mmol% (24 mg) of anhydrous $AlCl_3$ and one mmol (70 mg, 99%) of 2,3-Dihydrofuran was used. The resulting crude reaction mixture was purified by column chromatography using an AcOEt/Hexane mobile phase, yielding **CR80** in a 50% chemical yield.

### Solubility and HPLC method to CR80

Ten milligrams of CR80 were added to ten milliliters of phosphate-buffered saline (PBS) at a concentration of 1X and a pH of 7.35. The suspension was sonicated for one hour at room temperature. Subsequently, the resulting suspension was centrifuged at 4000 rpm for 20 min. Then, the resulting solution was filtered, and the solid was discarded. On the other hand, a method was developed: HPLC-RP with a flow of 1 mL/min at 235–250 nm, using a mobile phase of (30/70) water/ACN on a Shimadzu C18 50 × 4.6 mm column. The concentration of CR80 in PBS was established by interpolating the calibration curve of CR80, which exhibits a concentration range of 4–451 micromolar.

### Cell lines and culture conditions

The histiocytic lymphoma cell line U937 (ATCC® HTB-22TM) was used to determine the antitumor potential of the **CR80**. Additionally, the healthy cell line L929 was evaluated as a cytotoxicity control. The cell lines were cultured in DMEM (Dulbecco's modified Eagle) supplemented with 10% fetal bovine serum (FBS-Gibco, Fischer scientific, Alcobendas Madrid Spain). The cells were incubated in a humidified atmosphere with 5% $CO_2$ at 37°C. They were provided with fresh culture medium three times a week until they reached confluence. Adherent cells were harvested using a 0.25% trypsin-EDTA solution. On the other hand, for non-adherent U937 cells, RPMI 1640 medium with stable glutamine, 25 mM HEPES, and 10% FBS were used.

### Cytotoxicity screening

The Alamar blue Assay was used to determine the effect of **CR80** on tumor and healthy cells. Cells in 96-well microplates at a confluence of 10,000 cells per well were seeded. They were treated 24 hours after seeding with four concentrations of **CR80** (0.2, 0.4, 0.8, and 1.0 micromolar) and 7,9 micromolar value predicted by the QSAR model. The cytotoxic effect was assessed 24, 48, and 72 hours after treatment. The chemotherapeutic doxorubicin at 25 nM was used as a positive control, and untreated cells were used as a negative control. Briefly, the medium was replaced by 100 μL of Alamar blue reagent (40 μM), and the microplates were incubated for 4 hours under standard culture conditions and read the fluorescence in a microplate reader (530–590 nm, Tecan, Infinite® 200 PRO). Finally, the selectivity index (SI) of **CR80** was evaluated. The SI was calculated using the formula: SI = ($IC_{50}$ for normal cell line L-929)/ ($IC_{50}$ for U-937) [29]. A favorable SI > 1.0 indicates a drug with greater efficacy against tumor cells than toxicity against normal cells. All experiments were assessed in triplicate.

### Statistical analysis

Data were expressed as arithmetic mean ± SEM. Statistical analysis and graphical representation of the results were performed using GraphPad Prism software. Multiple comparisons were made between treatment concentrations and untreated cell viability. The Shapiro-Wilk test was conducted to determine data normality and ANOVA for multiple comparisons. A p-value less than 0.05 was considered statistically significant.

## Results and discussion

We identified over 150 anticancer molecules, with eight approved for treating acute lymphoblastic leukemia (ALL). Of these, 5 are small molecules: nelarabine (1), vincristine (2), etoposide (3), teniposide (4), and dactinomycin (5) (see Table 1).

Additionally, we found 30 molecules specifically targeting beta-tubulin, a promising target because leukemic cells, like those in ALL, divide more rapidly than normal cells. This rapid division can enhance beta-tubulin expression, making these cells more vulnerable to microtubule-interfering agents, such as vincas (vincristine) and taxanes. Therefore, molecules with greater affinity for beta-tubulin will selectively target cells with accelerated division, meaning treatments with a higher affinity for beta-tubulin preferentially affect leukemic cells over healthy cells due to the latter dividing slower [30–31]. We then used these 35 molecules for structure-based drug design (SBDD) and ligand-based drug design (LBDD). We started with LBDD and began by conducting a boxplot analysis, which led us to eliminate fourteen molecules based on their $IC_{50}$ values and poor selectivity. This left us with 21 molecules for further study. We began by selecting six molecular descriptors for input for the QSAR mathematical model. These descriptors were chosen through a screening process that reduced the initial 1,540 descriptors to 1,100 by eliminating those with zero values. Subsequently, we performed a descriptor vs. descriptor Pearson correlation, retaining only descriptors with correlation values between 0.2 and -0.2. We then conducted a new Pearson descriptor vs. $IC_{50}$ correlation using the filtered descriptors, aiming for correlation values close to one. From this analysis, we selected the six descriptors used to train the INQA-ANN. We conducted ten training sessions, adjusting the number of nodes until achieving a minimum Neural Network cost of $R^2=0.7$, which provided an accurate prediction function. The results are displayed in Graph 1a, showing an $R^2 = 0.734$ and including the model's cross-validation, demonstrating the validity of the built QSAR model. With a small amount of data but a strict selection, the model's noise is reduced, obtaining a model with high predictive capacity [32].

After constructing the QSAR model, we continued with SBDD. To do this, we calculated the affinity energies in kcal/mol for the twenty-one molecules used as a training set. We used the Beta-tubulin structure with the code 6QUS as the

**Table 1. Screening of commercial and experimental drugs for LBDD and SBDD.**

| CN | Affinity (Kcal/mol) | Log P (o/w) | $IC_{50}$ Experimental (μM) | $IC_{50}$ Predicted ANN (μM) | Toxicity[a] | | |
|---|---|---|---|---|---|---|---|
| | | | | | hERG Blockers | Ames Toxicity | Rat Oral Acute toxicity |
| 1 | −7.01 | −0.097 | 23.55 | 7.24 | 0.054 | 0.74 | 0.444 |
| 2 | −6.25 | 3.693 | 0.095 | 1.11 | 0.217 | 0.547 | 0.317 |
| 3 | −7.95 | 1.257 | 7.61 | 4.34 | 0.067 | 0.995 | 0.544 |
| 4 | −8,90 | 2.801 | 13.99 | 5.26 | 0.088 | 0.163 | 0.275 |
| 5 | −6.11 | 3.246 | 13.79 | 4.61 | 0.41 | 0.154 | 0.482 |
| 6 | −5.98 | 3.946 | 0.066 | 6.15 | 0.432 | 0.101 | 0.502 |
| 7 | −6.08 | 3.557 | 0.103 | 5.24 | 0.249 | 0.247 | 0.49 |
| 8 | −5.98 | 3.848 | 0.135 | 6.10 | 0.309 | 0.097 | 0.608 |
| 9 | −7.92 | 3.349 | 0.92 | 5.03 | 0.541 | 0.747 | 0.578 |
| 10 | −7.47 | 4.575 | 1 | 6.44 | 0.814 | 0.58 | 0.813 |
| 11 | −7.64 | 4.184 | 1 | 4.75 | 0.641 | 0.864 | 0.61 |
| 12 | −7.64 | 8.875 | 1 | 4.90 | 0.776 | 0.895 | 0.634 |
| 13 | −7.74 | 4.459 | 1 | 6.16 | 0.774 | 0.761 | 0.564 |
| 14 | −7.35 | 3.87 | 15 | 5.90 | 0.346 | 0.698 | 0.817 |
| 15 | −7.92 | 3.867 | 1.7 | 5.23 | 0.225 | 0.415 | 0.593 |
| 16 | −6.8 | 3.713 | 47.75 | 5.02 | 0.154 | 0.522 | 0.706 |
| 17 | −7.61 | 2.307 | 9.5 | 4.66 | 0.63 | 0.684 | 0.934 |
| 18 | −7.08 | 1.911 | 4.16 | 4.79 | 0.687 | 0.736 | 0.871 |
| 19 | −8.23 | 4.929 | 2.5 | 4.67 | 0.868 | 0.472 | 0.895 |
| 20 | −6.61 | 3.072 | 10 | 7.99 | 0.035 | 0.162 | 0.339 |
| 21 | −5.5 | 3.304 | 0.0029 | 8.00 | 0.054 | 0.075 | 0.054 |

[a]Calculated through software https://admetlab3.scbdd.com

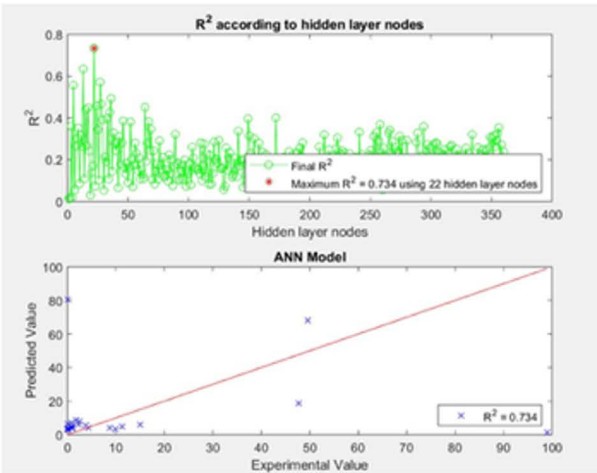

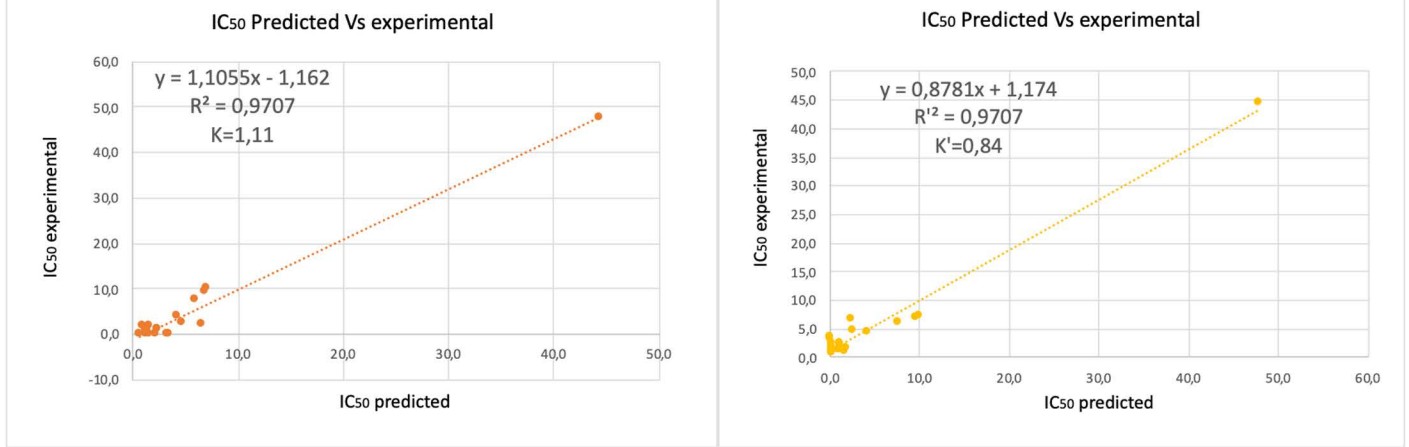

**Graph 1. a) R2 ANN model, b) external cross-validation.**

molecular target, obtained from the RCSB Protein Data Bank (RCSB PDB). This protein is a microtubule-organizing protein that specifically binds to the minus end of non-centrosome microtubules and regulates their dynamics and organization. Vincristine was used as a validation ligand with demonstrated activity and compared with the designed analogs (entry table 1 [31]. We additionally predict the $IC_{50}$ values and affinities for the molecular target and calculate the logP and toxicity values of the hit molecules using admelab 3.0 [33]. This helps us establish minimum quality criteria and design molecules that improve all pharmacological attributes evaluated in this set Table 1.

Upon analyzing the structures and results presented in Table 1, it was observed that entries 1–6 exhibit a structure analog to Carbamazepine (CBZ) (SI) [34]. CBZ is an approved drug used for treating epilepsy and pain related to trigeminal neuralgia. Modifying its core structure could enhance its selectivity over beta-tubulin and improve its pharmacodynamic and pharmacokinetic profile compared to other compounds with different structural groupings. To explore this, 57 structural changes were made, including homologous series, bioisosteric changes, and ring replacements in the nucleophilic and electrophilic positions of the CBZ nucleus (see Scheme 1).

The designed molecules' $IC_{50}$ value was predicted using the QSAR-ANN model constructed. Additionally, their affinity for the target beta-tubulin in kcal/mol was calculated, and both the logP value and toxicity were determined using AdmeLab software 3.0, as shown in Table 2.

22 R1=H, R2=H, R3=H
23 R1=CH$_3$, R2=H, R3=H
24 R1=Ethyl, R2=H, R3=H
25 R1=Propyl, R2=H, R3=H
26 R1=Butyl, R2=H, R3=H
27 R1=Pentyl, R2=H, R3=H
28 R1=NO$_2$, R2=H, R3=H
29 R1=Cl, R2=H, R3=H
30 R1=Br, R2=H, R3=H
31 R1=I, R2=H, R3=H
32 R1=F, R2=H, R3=H
33 R1=COCH$_3$, R2=H, R3=H
34 R1=COCH$_2$CH$_3$, R2=H, R3=H
35 R1=CO(CH$_2$)$_2$CH$_3$, R2=H, R3=H
36 R1=CO(CH$_2$)$_3$CH$_3$, R2=H, R3=H
37 R1=CO(CH$_2$)$_4$CH$_3$, R2=H, R3=H

38 R1= [4-chloroquinoline-7-yl] R2=H, R3=H
39 R1= [phenyl] R2=H, R3=H
40 R1= [cyclohexyl] R2=H, R3=H
41 R1= [cyclopentyl] R2=H, R3=H
42 R1= [pyrrolidinyl] R2=H, R3=H
43 R1= [tetrahydrofuranyl] R2=H, R3=H

44 R1=H, R2=CH$_3$, R3=H
45 R1=H, R2=Ethyl, R3=H
46 R1=H, R2=Propyl, R3=H
47 R1=H, R2=Butyl,, R3=H
48 R1=H, R2=Pentyl, R3=H
49 R1=H, R2=NO$_2$, R3=H
50 R1=H, R2=Cl, R3=H
51 R1=H, R2=Br, R3=H
52 R1=H, R2=I, R3=H
53 R1=H, R2=F, R3=H
54 R1=H, R2=COCH$_3$, R3=H
55 R1=H, R2=COCH$_2$CH$_3$, R3=H
56 R1=H, R2=CO(CH$_2$)$_2$CH$_3$, R3=H
57 R1=H R2=CO(CH$_2$)$_3$CH$_3$, R3=H
58 R1=H, R2=CO(CH$_2$)$_4$CH, R3=H

59 R2= [4-chloroquinoline-7-yl] R1=H, R3=H
60 R2= [phenyl] R1=H, R3=H
61 R2= [cyclohexyl] R1=H, R3=H
62 R2= [cyclopentyl] R1=H, R3=H
63 R2= [pyrrolidinyl] R1=H, R3=H
64 R2= [tetrahydrofuranyl] R1=H, R3=H

65 R1=H, R2=H, R3=CH$_3$
66 R1=H, R2=H, R3=Ethyl,
67 R1=H, R2=H, R3=Propyl
68 R1=H, R2=H, R3=Butyl
69 R1=H, R2=H, R3=Pentyl
70 R1=H, R2=H, R3=COCH$_3$
71 R1=H, R2=H,R3=COCH$_2$CH$_3$
72 R1=H, R2=H, R3=CO(CH$_2$)$_2$CH$_3$
73 R1=H R2=H, R3=CO(CH$_2$)$_3$CH$_3$
74 R1=H, R2H, =R3=CO(CH$_2$)$_4$CH,

75 R3= [4-chloro-7-methylquinolinyl] R1=H, R2=H
76 R3= [phenyl] R1=H, R2=H
77 R3= [cyclohexyl] R1=H, R2=H
78 R3= [cyclopentyl] R1=H, R2=H
79 R3= [pyrrolidinyl] R1=H, R2=H
80 R3= [tetrahydrofuranyl] R1=H, R2=H

**Scheme 1. Eighty structural changes to the CBZ nucleus.**

The criteria for selecting the designed molecules were as follows: the $IC_{50}$ values should be less than ten micromolar, the affinity for the molecular target should be more negative than -8 kcal/mol (keeping in mind that the CBZ nucleus presents a value of −7 kcal/mol), and the logP values should fall within the range of 2–4, as this range has experimentally shown promising results. Ultimately, the toxicity profile must match or surpass that of current alternatives drugs.

When analyzing the predicted values, 90% of the candidates exceed the $IC_{50}$ filter of less than ten micromolar Table 2. This result can be attributed to the fact that the CBZ nucleus is conserved in all the designed analogs, and the modifications made are auxophoric. This means the changes do not significantly vary the biological activity value but make them good candidates. However, these auxophoric modifications substantially impact the candidate's pharmacodynamics. The results obtained from calculating the affinity in kcal/mol against the beta-tubulin target found a range from −6.5 kcal/

**Table 2. LBVS and SBVS results for CBZ and CBZ analogs.**

| CR | Affinity (Kcal/mol) | Log P (o/w) | IC₅₀ Predicted ANN (µM) | Toxicity[a] | | | CR | Affinity (Kcal/mol) | Log P (o/w) | IC₅₀ Predicted ANN (µM) | Toxicity[a] | | |
|----|----|----|----|----|----|----|----|----|----|----|----|----|----|
| | | | | hERG Blockers | Ames Toxicity | Rat Oral Acute toxicity | | | | | hERG Blockers | Ames Toxicity | Rat Oral Acute toxicity |
| 22 | −7.04 | 2.357 | 8.1 | 0.244 | 0.822 | 0.344 | 52 | −6.97 | 2.584 | 8.2 | 0.185 | 0.869 | 0.349 |
| 23 | −7.56 | 2.692 | 7.9 | 0.268 | 0.817 | 0.306 | 53 | −8.14 | 1.678 | 7.9 | 0.239 | 0.692 | 0.362 |
| 24 | −6.94 | 3.314 | 7.9 | 0.326 | 0.847 | 0.193 | 54 | −7.58 | 1.769 | 8.0 | 0.145 | 0.718 | 0.269 |
| 25 | −7.09 | 3.666 | 8.0 | 0.498 | 0.739 | 0.234 | 55 | −7.43 | 1.879 | 8.2 | 0.201 | 0.707 | 0.314 |
| 26 | −7.16 | 4.285 | 8.2 | 0.671 | 0.625 | 0.247 | 56 | −7.1 | 2.254 | 7.9 | 0.204 | 0.528 | 0.241 |
| 27 | −7.12 | 4.703 | 7.9 | 0.806 | 0.607 | 0.296 | 57 | −7.03 | 3.076 | 8.1 | 0.262 | 0.376 | 0.224 |
| 28 | −7.62 | 2.338 | 7.9 | 0.378 | 0.979 | 0.517 | 58 | −6.9 | 3.548 | 8.1 | 0.364 | 0.367 | 0.204 |
| 29 | −7.13 | 2.981 | 8.0 | 0.382 | 0.735 | 0.371 | 59 | −8.95 | 3.693 | 8.2 | 0.567 | 0.842 | 0.661 |
| 30 | −7.04 | 3.025 | 8.0 | 0.283 | 0.68 | 0.438 | 60 | −7.81 | 3.049 | 7.9 | 0.4 | 0.762 | 0.393 |
| 31 | −6.8 | 3.005 | 8.2 | 0.712 | 0.42 | 0.406 | 61 | −7.94 | 3.631 | 8.2 | 0.378 | 0.695 | 0.206 |
| 32 | −7.11 | 2.525 | 7.9 | 0.333 | 0.861 | 0.534 | 62 | −7.35 | 3.34 | 8.1 | 0.342 | 0.763 | 0.245 |
| 33 | −7.55 | 2.584 | 8.1 | 0.196 | 0.651 | 0.479 | 63 | −7.31 | 1.911 | 8.0 | 0.419 | 0.739 | 0.493 |
| 34 | −7.62 | 2.04 | 8.1 | 0.328 | 0.907 | 0.328 | 64 | −7.45 | 2.278 | 8.2 | 0.26 | 0.758 | 0.363 |
| 35 | −7.45 | 2.784 | **8.2** | 0.351 | 0.899 | 0.551 | 65 | −6.75 | 2.89 | 7.9 | 0.163 | 0.674 | 0.414 |
| 36 | −7.43 | 3.068 | 7.9 | 0.383 | 0.868 | 0.515 | 66 | −6.86 | 3.27 | 8.2 | 0.199 | 0.689 | 0.331 |
| 37 | −7.29 | 3.553 | 8.2 | 0.498 | 0.838 | 0.523 | 67 | −6.68 | 3.902 | 8.1 | 0.285 | 0.574 | 0.344 |
| 38 | −9.54 | 4.344 | 8.1 | 0.814 | 0.947 | 0.647 | 68 | −6.75 | 4.006 | 7.9 | 0.199 | 0.637 | 0.351 |
| 39 | −7.91 | 3.443 | 8.0 | 0.624 | 0.901 | 0.49 | 69 | −6.83 | 4.468 | 8.2 | 0.269 | 0.571 | 0.398 |
| 40 | −8.33 | 4.288 | 8.2 | 0.567 | 0.832 | 0.526 | 70 | −7.78 | 2.84 | 8.1 | 0.065 | 0.491 | 0.274 |
| 41 | −7.37 | 3.932 | 8.0 | 0.552 | 0.86 | 0.587 | 71 | −7.87 | 3.025 | 8.0 | 0.186 | 0.544 | 0.372 |
| 42 | −7.41 | 2.039 | 8.2 | 0.623 | 0.848 | 0.75 | 72 | −7.91 | 3.606 | 7.9 | 0.188 | 0.48 | 0.338 |
| 43 | −7.63 | 2.66 | 7.9 | 0.466 | 0.878 | 0.593 | 73 | −3,60 | 4.123 | 8.2 | 0.268 | 0.373 | 0.356 |
| 44 | −7.15 | 2.183 | 8.1 | 0.22 | 0.795 | 0.307 | 74 | −3,59 | 4.588 | 8.1 | 0.346 | 0.503 | 0.349 |
| 45 | −6.9 | 2.801 | 8.1 | 0.247 | 0.681 | 0.27 | 75 | −8.83 | 5.242 | 8.0 | 0.521 | 0.747 | 0.515 |
| 46 | −6.8 | 3.28 | **8.2** | 0.284 | 0.675 | 0.235 | 76 | −7.87 | 4.157 | 8.0 | 0.393 | 0.694 | 0.324 |
| 47 | −7.1 | 3.762 | 7.9 | 0.361 | 0.606 | 0.251 | 77 | −7.86 | 4.318 | 7.9 | 0.29 | 0.494 | 0.258 |
| 48 | −6.97 | 4.126 | 8.2 | 0.391 | 0.571 | 0.256 | 78 | −7.47 | 3.412 | 8.2 | 0.213 | 0.675 | 0.413 |
| 49 | −7.56 | 1.406 | 8.1 | 0.109 | 0.955 | 0.415 | 79 | −7.6 | 4.031 | 8.0 | 0.275 | 0.545 | 0.303 |
| 50 | −7.3 | 2.044 | 8.0 | 0.217 | 0.81 | 0.313 | 80 | **−8.66** | **3.415** | **7.8** | **0.377** | **0.552** | **0.601** |
| 51 | −7.39 | 2.308 | 8.2 | 0.125 | 0.862 | 0.48 | | | | | | | |

[a]Calculated through software https://admetlab3.scbdd.com

mol to −9.5 kcal/mol, demonstrating a significant difference. This variance indicates that the hydrophobic interactions between beta-tubulin and the analogs increase the affinity. Only six candidates designed exceed the -8 kcal/mol affinity value established as a quality criterion. Of these six candidates (entries 38, 40, 53, 59, 75, and 80 Table 2), three have LogP values in the promising range of logP 2–4 (entries 53, 59, 80, Table 2), While the remaining three are outside the established range, these candidates have security profiles equivalent to CBZ and commercial drugs. However, candidate 80 has the best toxicity profile, better than the three selected candidates so that it will proceed to the synthesis stage Scheme 2.

### Synthesis of N-(2-tetrahydrofuran)-Carbamazepine and characterization CR80

Scheme 3 describes the reaction between 2,3-Dihydrofuran (DHF) **2** and CBZ **1**, with aluminum trichloride as the catalyst. Some researchers have proposed methods for obtaining N-substituted amides [35–37], but we have our approach. We optimized the reaction conditions by experimenting with different solvents, catalyst proportions, and stoichiometric ratios. The best conditions we found were using ten mol% of anhydrous aluminum trichloride, a 1:2 CBZ to DHF ratio, and refluxing in THF for three hours. This resulted in a 50% yield of **CR80** (N-(2-tetrahydrofuran)-5H-dibenzo[b,f]azepine-5-carboxamide). We purified the compound using a mixture of ethyl acetate and hexane in a 2:1 ratio. $^{1}$H-NMR (400 MHz, Chloroform-$d$) δ 7.39 (ddt, $J$ = 39.1, 12.3, 7.9 Hz, 8H), 6.92 (d, $J$ = 1.9 Hz, 2H), 5.62–5.52 (m, 1H), 4.72 (d, $J$ = 8.4 Hz, 1H), 3.76 (dd, $J$ = 28.6, 7.3 Hz, 2H), 2.11–2.04 (m, 1H), 1.79 (dt, $J$ = 13.1, 6.8 Hz, 2H), 1.48 (d, $J$ = 7.0 Hz, 1H). $^{13}$C-NMR (100 MHz, Chloroform-$d$) 156.8, 137.4 137.2 132, 129.4, 129.3 128.4, 128.2, 127.5, 127,4, 118.1 118.0, 68, 67,9, 26, 25,9 FT-IR (neat) u(cm$^{-1}$): 3342, 3062, 3020, 2955, 2928, 1708, 1345, 1290, 1133, 1101. HRMS (ESI): $C_{19}H_{19}N_2O_2$ [M + H$^+$]: calc. 307.1447, found. 307.14529. DRX characterization (S7)

The biological activity of the synthesized **CR80** compound was evaluated *in vitro*. A calibration curve was constructed using **CR80** dissolved in ethyl acetate. The area under the curve of a **CR80** solution prepared in PBS (1X pH 7.35) was interpolated, yielding a concentration of 10 micromolar. We analyzed the cytotoxic effect on the human lymphoma-derived tumor cell line U-937 and the healthy fibroblast line L-929 to assess the selectivity of the synthesized drug, the cytotoxic

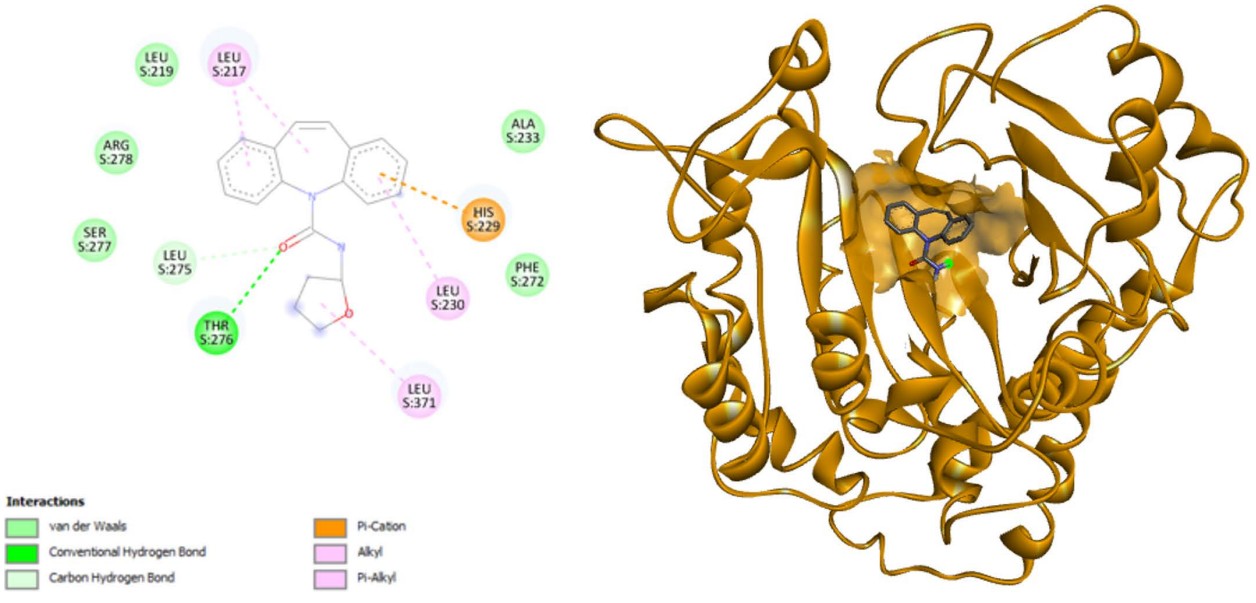

**Scheme 2. Non-covalent interactions CR80 and 3D representation.**

**Scheme 3. Reaction equation for the synthesis of CR80.**

effect of CR80 on the U937 and L929 cell lines was evaluated at a concentration of 7.9 micromolar, as predicted by the neuronal network (refer to Table 2, entry 80). This concentration caused cytotoxicity in the U937 and healthy L929 lines, resulting in a non-viable selectivity index (data excluded). It is crucial to understand that the developed prediction function does not support forecasting the SI selectivity index, as it relies solely on the $IC_{50}$ values of cancer cells; this represents a methodological limitation we must consider. Although CR80 showed promising activity, its low safety profile led us to conduct dilutions in the nanomolar range, concentrating on four levels: 0.2, 0.4, 0.8, and 1 micromolar. Doxorubicin was used as a positive control at 25 nM. Cytotoxicity was assessed at 24, 48, and 72 hours, but at 24 hours, the cytotoxic effect was not apparent (data not shown). However, at 48 hours (Fig 1A), statistically significant differences were observed at 0.8 and 1 micromolar compared to the untreated control, showing a better cytotoxic effect and more excellent safety in the healthy cell line evaluated.

Similarly, after the evaluation at 72 hours (Fig 1B), a decrease in the percentage of cell viability close to 50% was evident at all concentrations, maintaining a safety profile at 0.2, 0.4, and 0.8 micromolar compared to the untreated control with most statistics differences. Additionally, it showed better results in reducing cell viability than chemotherapeutic Doxorubicin treatment. The SI obtained was equal to 2, showing a highly favorable result regarding the selectivity of the synthesized molecule against the evaluated tumor line, instilling optimism about its potential.

We confirmed that the *in vitro* values predicted by the QSAR model developed with the INQA-ANN align when assessing the U -937 cell line. However, its impact on healthy cells (L929) shows toxicity at 10 micromolar. To improve the selectivity index (SI), we performed dilutions that yielded excellent SI results. This demonstrates that the synergistic approach combining LBDD and SBDD can lead to identifying a promising new drug for *in vivo* trials. Furthermore, we are developing new QSAR models to predict the activity against cancer cells and the selection index, which will serve as a crucial criterion for new lead compounds.

## Conclusions

A new candidate has been developed rationally for potentially treating Acute Lymphoblastic Leukemia. It was designed through the synergy between ligand-based and structure-based drug design. The starting point was the pharmacophoric core of carbamazepine, which underwent more than fifty modifications to obtain an analog with greater affinity for the target beta-tubulin (−8.66 Kcal/mol). This candidate has shown a promising $IC_{50}$ 0.8–1 micromolar value *in vitro* on the U-937

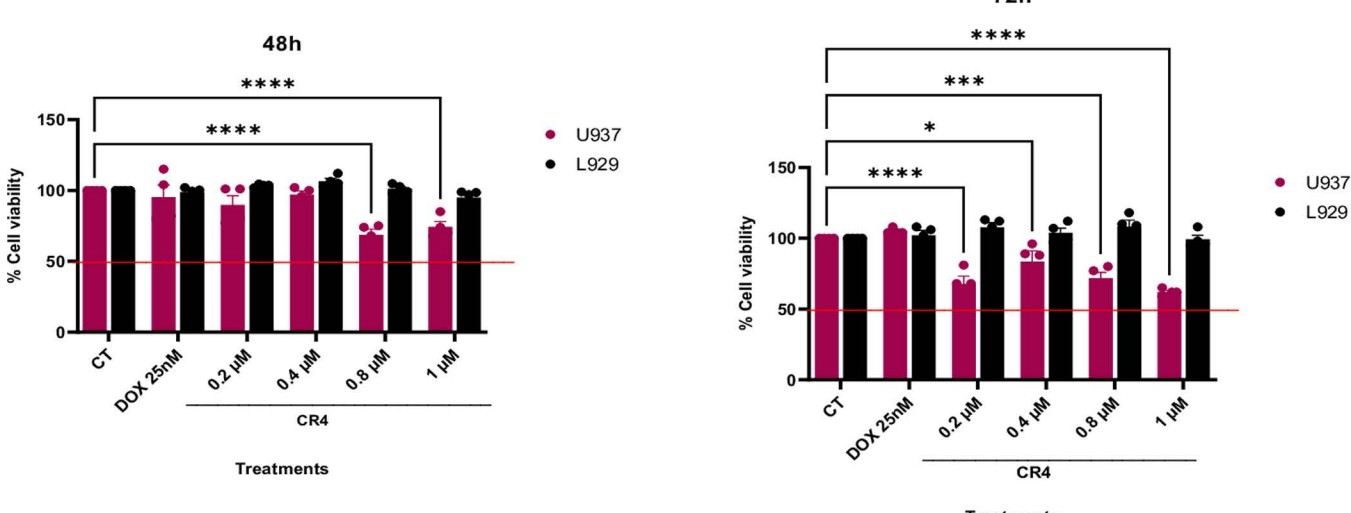

**Fig 1. Cytotoxicity concentration-response curves for CR80.** U-937 and L-929 cell lines were treated with CR80 (CR4 into, fig 1), and cytotoxic effects were assessed at (A) 48 hours and (B) 72 hours. Statistical significance is indicated as follows: () p < 0.05, (**) p < 0.005, and (****) p < 0.0005. All experiments were assessed in triplicate.

cell line and an SI of 2, making it a promising candidate for *in vivo* trials. This development continues to prove the importance of rational computer-assisted drug design.

## Supporting information

**S1 and S2 Fig. Pearson Correlation.**
(DOCX)

**S2 File. CR80 characterization.**
(DOCX)

**S3 Table. Table 1SI and Table 2SI.**
(DOCX)

## Acknowledgments

I want to thank the INQA group and the pharmaceutical chemistry program for their support at Universidad El Bosque in Bogotá, Colombia.

## Author contributions

**Conceptualization:** Angela V. Fonseca-Benítez, James Guevara Pulido.

**Data curation:** Angela V. Fonseca-Benítez.

**Formal analysis:** Angela V. Fonseca-Benítez, James Guevara Pulido.

**Funding acquisition:** James Guevara Pulido.

**Investigation:** Cristian Álvarez-Gómez.

**Methodology:** Cristian Álvarez-Gómez.

**Project administration:** James Guevara Pulido.

**Resources:** James Guevara Pulido.

**Software:** Cristian Álvarez-Gómez.

**Validation:** James Guevara Pulido.

**Visualization:** James Guevara Pulido.

**Writing – original draft:** Angela V. Fonseca-Benítez, James Guevara Pulido.

**Writing – review & editing:** James Guevara Pulido.

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
