## [Decision Letter · Decision Letter 0]

15 Oct 2024

PONE-D-24-38959Design, Synthesis, and In Vitro Evaluation of a Carbamazepine Derivative with Antitumor Potential in a Model of Acute Lymphoblastic LeukemiaPLOS ONE

Dear Dr. Guevara Pulido,

Thank you for submitting your manuscript to PLOS ONE. After careful consideration, we feel that it has merit but does not fully meet PLOS ONE’s publication criteria as it currently stands. Therefore, we invite you to submit a revised version of the manuscript that addresses the points raised during the review process.

We look forward to receiving your revised manuscript.

Kind regards,

Sapan Kamleshkumar Shah, Ph.D., M.Pharm.

Academic Editor

PLOS ONE

“This project was funded by the INQA Research Group, which is associated with the Pharmaceutical Chemistry Program at Universidad El Bosque, Bogotá, Colombia.”

Additional Editor Comments:

Abstract need to be refurbish. Many grammatical mistakes are provided in abstract.

At least 5 key words should be added by authors.

The statement “The affinity of ALL and beta-tubulin drugs and their designed analogs was evaluated in kcal/mol using the PDB 6QUS crystal structure” not justified?

Sentence never start with value eg. “10 mg of CR80 was added to 10 mL of (PBS, 1X pH 7.35)” grammatically incorrect.

What is meaning of this “CR80 dissolved in Ethyl Acetate 451.2;121.2; y 41.01 µM.??”

Overall, authors utilized only 21 molecules for QSAR model building. These no. is too less for building statistically robust QSAR. Further authors have not validated QSAR model according to OECD guidelines? So, applicability of build model for screening external dataset is doubtful and meaningless as authors have not even performed splitting of dataset, internal validation and Applicability domain studies?

Authors have performed molecular docking studies on selected 21 molecules. However, they have not performed any cross validation studies of docking results even with standard inhibitors of beta tubulin or native inhibitors of 6QUS taxol? How authors will confirmed surety of docking studies? Howe molecular docking results are helpful for authors they have not mentioned any significance of it? Authors used both approach LBDD and SBDD but not correlated both approached design?

Authors have not given detail account on design of 57 compounds using various methods. Authors must give details and elaborate on this design? Is selected 6 descriptors in QSAR models, Molecular docking interaction how contribute for these 57 compound design?

Authors have designed 57 compounds and selected those compound having IC50 value less that 10 micro molar. What is the logic for selecting such large value? Generally compound with less than 1 micromolar value are rational to select?

Authors claim that CR80 less toxic. However, as per my knowledge compounds with statistical values in table above 0.5 for HERG, AMES and RAT TOXICITY are not good and in table CR80 has AMES 0.552 and RAT toxicity 0.601 which is quite high? Authors claim CR80 has best toxicity profile, however, one can see from table 2 compounds 57,58,70, 73, 77 and even 79 have better profile over CR80?

Authors have predicted IC50 value of compound CR80 to be 8.1 mircomolar. However, while performing invitro analysis authors used concentration only upto 1000 nanomolar? What its rationale for these dosage?

Authors have provided so many studies but relevancy, validation and accuracy of data is not justified. I would suggest major revision to authors for above points including reviewer comments.

Reviewers' comments:

Reviewer's Responses to Questions

**Comments to the Author**

1. Is the manuscript technically sound, and do the data support the conclusions?

Reviewer #1: Yes

Reviewer #2: Yes

2. Has the statistical analysis been performed appropriately and rigorously? 

Reviewer #1: Yes

Reviewer #2: No

3. Have the authors made all data underlying the findings in their manuscript fully available?

Reviewer #1: Yes

Reviewer #2: Yes

4. Is the manuscript presented in an intelligible fashion and written in standard English?

Reviewer #1: Yes

Reviewer #2: Yes

5. Review Comments to the Author

Reviewer #1: The manuscript is well written. Authors use In-silico approaches for selection and designing of molecule, Also, synthesis and in vitro experiments were performed. However, some additional detail are needed to enhance the its impact

In introduction, brief overview of the current therapies for acute lymphoblastic leukemia should be include. Also, rationale for selecting carbamazepine as a lead compound should be explain.

Some more details on the molecular docking and software used for binding affinity predictions would strengthen the methodology section.

In discussion, how findings of studies correlate with potential mechanisms of CR80 through beta-tubulin should include.

Reviewer #2: 1. Please elaborate on the role of target selected i.e. beta tubulin in ALL

2. It is not stated anywhere how the data is recorded for in vitro test... duplicate or triplicate ?

3. There is no mention of probability value for toxicity value output... so it not clear

4. Rationale behind the selection of CR80 is not satisfactory as the similar values are observed for other derivatives

6. PLOS authors have the option to publish the peer review history of their article (what does this mean? ). If published, this will include your full peer review and any attached files.

**Do you want your identity to be public for this peer review?** For information about this choice, including consent withdrawal, please see our Privacy Policy .

Reviewer #1: No

Reviewer #2: No

---

## [Author Response · Author response to Decision Letter 1]

30 Oct 2024

Additional Editor Comments:

Abstract need to be refurbish. Many grammatical mistakes are provided in abstract.

• Response: Thanks for the correction. All grammatical errors were corrected.

Acute lymphoblastic leukemia (ALL) is a significant concern in both pediatric and adult demographics. Despite 156 approved cancer therapies based on small molecules, a mere five apply to all types of leukemia. Unfortunately, adherence to these treatments is low due to adverse side effects. Consequently, there is an urgent need to identify more effective treatment options for ALL. This study presents a potential solution. We have designed over fifty analogs of carbamazepine, utilizing a combination of ligand-based and structure-based drug design methodologies. Among these analogs, we identified the CR80 analog, which demonstrated predicted binding values of -8.66 kcal/mol against beta-tubulin, a favorable LogP, and IC50 values suitable for in vitro evaluation. The CR80 compound was synthesized with a yield of 50% and subsequently assessed in vitro against the U-937 cell line. It obtained an IC50 value of 800 nM to 1000 nM and a selectivity index of two, thus marking it as a promising candidate for in vivo studies.

At least 5 key words should be added by authors.

• Response: Beta-tubulin, drug discovery, and in vitro were added as keywords

The statement “The affinity of ALL and beta-tubulin drugs and their designed analogs was evaluated in kcal/mol using the PDB 6QUS crystal structure” not justified?

• Response: thank you for the comment. Indeed, we had not explained this statement in the best way. It was corrected as follows.

“The affinity of drugs with pharmacological activity against beta-tubulin, their designed analogs, and other drugs commonly used for the treatment of ALL was evaluated in kcal/mol using the PDB 6QUS crystal structure.

Sentence never start with value eg. “10 mg of CR80 was added to 10 mL of (PBS, 1X pH 7.35)” grammatically incorrect.

• Response: Thank you for the correction.

“Ten milligrams of CR80 were added to ten milliliters of phosphate-buffered saline (PBS) at a concentration of 1X and a pH of 7.35.”

What is meaning of this “CR80 dissolved in Ethyl Acetate 451.2;121.2; y 41.01 µM.??”

• Response: We wanted to say that the concentration of CR80 in PBS was established by interpolating the calibration curve of CR80, which exhibits a concentration range of 41 to 451 micromolar. This was corrected in the manuscript.

Overall, authors utilized only 21 molecules for QSAR model building. These no. is too less for building statistically robust QSAR. Further authors have not validated QSAR model according to OECD guidelines? So, applicability of build model for screening external dataset is doubtful and meaningless as authors have not even performed splitting of dataset, internal validation and Applicability domain studies?

• Response: In agreement with the comments, we allow ourselves to give the following explanation. In the most recent review published in July 2024 [1] the authors describe how a small number of data with a correct choice can offer a robust QSAR model “Despite the need for large training datasets being the 'Achilles' heel' of deep learning in drug discovery, several advances allow neural networks to be – paradoxically – powerful tools in low-data scenarios. An increasing body of literature shows how strategies like the ones discussed in this minireview can lead to high-performing deep learning models, even with little data.” In our QSAR model we reduce the noise of the model by carrying out an in-depth bibliographic review and grouping molecules with similar electronic and steric properties with a data normalization that allows a robust model verified by the internal validation of the model R2=0.734 and an external validation that exceeds the parameters described in the literature as described in graph 1.

For greater clarity, the following paragraph has been added in the document.

The results are displayed in graph 1a, showing an R² = 0.734 and including the model's cross-validation, demonstrating the validity of the built QSAR model. With a small amount of data but a strict selection, the model's noise is reduced, obtaining a model with high predictive capacity [2].

[1]Tilborg, D., Brinkmann, H., Criscuolo, E., Rossen, L., Özçelik, R., & Grisoni, F. (2024). Deep learning for low-data drug discovery: hurdles and opportunities. Current Opinion in Structural Biology, 86, 102818.

[2] Golbraikh, A., & Tropsha, A. (2000). Predictive QSAR modeling based on diversity sampling of experimental datasets for the training and test set selection. Molecular diversity, 5, 231-243.

Authors have performed molecular docking studies on selected 21 molecules. However, they have not performed any cross validation studies of docking results even with standard inhibitors of beta tubulin or native inhibitors of 6QUS taxol?

• Response: According to your question the docking was validated by docking with the ligand with activity demonstrated by beta tubulin vincristine [3] the energy was evaluated, and based on it the comparison was made as described in table 1.

The document was clarified for readers.

[3] Reichle, A., Diddens, H., Altmayr, F., Rastetter, J., & Andreesen, R. (1995). Beta-tubulin and P-glycoprotein: major determinants of vincristine accumulation in B-CLL cells. Leukemia research, 19(11), 823-829.

Authors used both approach LBDD and SBDD but not correlated both approached design?

• Response. According to your question, we would like to clarify that of course, we correlate both models; the selection criteria for our candidates are the following as described in the manuscript,

1. Affinity by Docking

2. IC50 By QSAR model

3. ADME evaluation

We additionally predict the IC50 values and affinities for the molecular target and calculate the logP and toxicity values of the hit molecules using admelab 3.0 [26]. This helps us establish minimum quality criteria and design molecules that improve all pharmacological attributes evaluated in this set Table 1

The criteria for selecting the designed molecules were as follows: the IC50 values should be less than ten micromolar, the affinity for the molecular target should be more negative than -8 kcal/mol (keeping in mind that the CBZ nucleus presents a value of -7 kcal/mol), and the logP values should fall within the range of 2 to 4, as this range has experimentally shown promising results. Finally, the toxicity profile should be equivalent to or better than that of existing drugs.

The authors have not given a detailed accounts of the design of 57 compounds using various methods. Authors must give details and elaborate on this design?

• Response: According to your question, in this paragraph described in the manuscript we establish the criteria for the design of the molecules.

“To explore this, 57 structural changes were made, including homologous series, bioisosteric changes, and ring replacements in the nucleophilic and electrophilic positions of the CBZ nucleus.”

This is done by seeking to optimize the subsequent synthesis process using a commercial building block such as CBZ for subsequent optimal scaling.

Is selected 6 descriptors in QSAR models, Molecular docking interaction how contribute for these 57 compound design?

• Response: Thank you for your question. The descriptors contribute to the pharmacophoric core of carbamazepine. Our 57 changes were made to the auxophores of carbamazepine, which kept the pharmacophoric grouping intact, ensuring biological activity but improving its pharmacophoric and dynamic attributes. It was added to the text pharmacophoric and dynamic attributes.

Authors have designed 57 compounds and selected those compound having IC50 value less that 10 micro molar. What is the logic for selecting such a large value? Generally compound with less than 1 micromolar value are rational to select?

• Response: Thanks for the question. If we are clear, values of 1 micromolar are ideal for rational design. However, the auxophoric changes generated predictions close to 8 micromolar, but since our only criterion is not the IC50 value, we consider that a synergy between the logP affinity and the activity value made the candidate promising, as was demonstrated in the in-house tests. vitro.

The authors claim that CR80 is less toxic. However, to my knowledge, compounds with statistical values above 0.5 for HERG, AMES, and RAT TOXICITY are not good. In table CR80, AMES is 0.552, and RAT toxicity is 0.601, which is quite high. The authors claim that CR80 has the best toxicity profile. However, one can see from table 2 that compounds 57, 58, 70, 73, 77, and even 79 have better profiles than CR80.

• Response: Thank you for your comment, we would like to clarify why the selection of CR80

The choice of the candidate did not depend exclusively on one or another criterion, our proposal is that with respect to Hit CBZ the pharmacodynamic and pharmacokinetic attributes would be improved; in this case, candidates 57, 58, 73, and 79 have log values greater than 4 lo which makes them disposable and 70 does not exceed the affinity, for that reason in this coma we do not select the mentioned candidates.

For readers' clarity, the paragraph was written as follows: While the remaining three are outside the established range, these candidates have security profiles equivalent to CBZ and commercial drugs. However, candidate 80 has the best toxicity profile, better than the three selected candidates, so it will proceed to the synthesis stage.

Authors have predicted IC50 value of compound CR80 to be 8.1 mircomolar. However, while performing invitro analysis authors used concentration only upto 1000 nanomolar? What its rationale for these dosage?

• Response: Thank you for the question. It clarified why the reported results are lower than those predicted by the neural network. Therefore, we allow ourselves to clarify the following in the document.

We analyzed the cytotoxic effect on the human lymphoma-derived tumor cell line U-937 and the healthy fibroblast line L-929 to assess the selectivity of the synthesized drug, the cytotoxic effect of CR80 on the U937 and L929 cell lines was evaluated at a concentration of 7.9 micromolar, as predicted by the neuronal network (refer to Table 2, entry 80). This concentration caused cytotoxicity in both the U937 and healthy L929 lines, resulting in a non-viable selectivity index (data not included). It is crucial to understand that the developed prediction function does not support forecasting the SI selectivity index, as it relies solely on the IC50 values of cancer cells; this represents a methodological limitation we must consider. Although CR80 showed promising activity, its low safety profile led us to conduct dilutions in the nanomolar range, concentrating on four levels: 200, 400, 800, and 1000 nM. Doxorubicin was used as a positive control at 25 nM.

We confirmed that the in vitro values predicted by the QSAR model developed with the INQA-ANN align when assessing the U -937 cell line. However, its impact on healthy cells, such as L929, shows toxicity at 10 micromolar. To improve the selectivity index (SI), we performed dilutions that yielded excellent SI results. This demonstrates that the synergistic approach combining LBDD and SBDD can lead to the identification of a promising new drug for in vivo trials. Furthermore, we are developing new QSAR models aimed at predicting both the activity against cancer cells and the selection index, which will serve as a crucial criterion for new lead compounds.

Reviewer #1: The manuscript is well written. Authors use In-silico approaches for selection and designing of molecule, Also, synthesis and in vitro experiments were performed. However, some additional detail are needed to enhance the its impact

In introduction, brief overview of the current therapies for acute lymphoblastic leukemia should be include. Also, rationale for selecting carbamazepine as a lead compound should be explain.

• Response: Thank you for your suggestion. Current therapies for the treatment of children and adults with ALL were included, and the manuscript was updated with ALL were included and the manuscript was updated.

The prognosis for patients with acute lymphoblastic leukemia (ALL) has greatly improved due to intensive multimodal treatment strategies, such as chemotherapy, high-dose chemotherapy with stem cell rescue, and radiation therapy when necessary [3]. The treatment of Adult Acute Lymphoblastic Leukemia (ALL) involves complex chemotherapy combinations and schedules typically seen in oncology. Two main chemotherapy regimens are currently used. The Berlin-Frankfurt-Münster protocol features an induction regimen, consolidation regimen, reintensification regimen, and maintenance therapy, primarily implemented in European adult ALL clinical trials. Alternatively, the hyper-CVAD regimen, created by MD Anderson Cancer Center researchers, consists of rotating two different intensive chemotherapy cycles [4].

3 Chang, J. H. C., Poppe, M. M., Hua, C. H., Marcus, K. J., & Esiashvili, N. (2021). Acute lymphoblastic leukemia. Pediatric Blood & Cancer, 68, e28371.

4 Imai, K. (2017). Acute lymphoblastic leukemia: pathophysiology and current therapy. [Rinsho Ketsueki] The Japanese Journal of Clinical Hematology, 58(5), 460-470.

Some more details on the molecular docking and software used for binding affinity predictions would strengthen the methodology section.

• Response: Thank you for your suggestion

The methodology was expanded in the manuscript as follows “The binding affinities of drugs that affect beta-tubulin, including their designed analogs and other frequently prescribed medications for treating ALL, were evaluated in kcal/mol utilizing the PDB 6QUS crystal structure [18]. Protein preparation followed the AutoDockTools protocol [19]. The co-crystallized ligand, paclitaxel, was removed using Samson software. After preparing the crystal, docking was conducted with the known active ligand, vincristine. Following validation of the docking, additional energies were calculated. The structures were modeled, and their energies were optimized in Avogadro [20] using the MMFF94s force field. Subsequently, 35 drugs and 58 designed analogs were docked with 6QUS. The grid box was set at 13 × 15 × 25 points with a grid spacing of 0.375 Å, centered at coordinates 2, 23, and 2. Calculations were conducted in triplicate, and the affinity energy of the pose with the lowest RMSD value was averaged for each compound. The interactions and distances were visualized using Discovery Studio Suite®.”

In discussion, how findings of studies correlate with potential mechanisms of CR80 through beta-tubulin should include.

• Response Thank you for your comments, in the manuscript we expand the role of ALL.

Additionally, we found 30 molecules specifically targeting beta-tubulin, a promising target because leukemic cells, like those in ALL, divide more rapidly than normal cells. This rapid division can enhance beta-tubulin expression, making these cells more vulnerable to microtubule-interfering agents, such as vincas (vincristine) and taxanes. Therefore, molecules with greater affinity for beta-tubulin will selectively target cells with accelerated division, meaning treatments with a higher affinity for beta-tubulin preferentially affect leukemic cells over healthy cells due to the latter dividing slower [X]

González-García, J. R., et al. (2019). "Targeting microtubules in cancer therapy." Cancer Treatment Reviews, 80, 101895.

Wang, X., Gigant, B., Zheng, X., & Chen, Q. (2023). Microtubule‐targeting agents for cancer treatment: Seven binding sites and three strategies. MedComm–Oncology, 2(3), e46.

Reviewer #2:

1. Please elaborate on the role of target selected i.e. beta tubulin in ALL

Response Thank you for your comments, in the manuscript we expand the role of ALL

Additionally, we found 30 molecules specifically targeting beta-tubulin, a promising target because leukemic cells, like those in ALL, divide more rapidly than normal cells. This rapid division can enhance beta-tubulin expression, making these cells more vulnerable to microtubule-interfering agents, such as vincas (vincristine) and taxanes. Therefore, molecules with

---

## [Decision Letter · Decision Letter 1]

14 Jan 2025

PONE-D-24-38959R1Design, Synthesis, and In Vitro Evaluation of a Carbamazepine Derivative with Antitumor Potential in a Model of Acute Lymphoblastic LeukemiaPLOS ONE

Dear Dr. Guevara Pulido,

Thank you for submitting your manuscript to PLOS ONE. After careful consideration, we feel that it has merit but does not fully meet PLOS ONE’s publication criteria as it currently stands. Therefore, we invite you to submit a revised version of the manuscript that addresses the points raised during the review process.

We look forward to receiving your revised manuscript.

Kind regards,

Sapan Kamleshkumar Shah, Ph.D., M.Pharm.

Academic Editor

PLOS ONE

Journal Requirements:

Additional Editor Comments:

Manuscript is ready to accept and accepted after some minor corrections.

1. As author said in the abstract that 156 approved cancer therapies based on small molecules are there so they should give reference for this.

2. The IC 50 values are in nanomolar and some in millimolar, authors should explain why they are different. Authors must use only one unit for all the compounds.

3. Authors should give numbering to the structures of scheme.

4. The word in vivo, in vitro, in silico must be in italics.

5. Authors must give graphs of IR, NMR and Mass of CR80.

Reviewers' comments:

Reviewer's Responses to Questions

**Comments to the Author**

1. If the authors have adequately addressed your comments raised in a previous round of review and you feel that this manuscript is now acceptable for publication, you may indicate that here to bypass the “Comments to the Author” section, enter your conflict of interest statement in the “Confidential to Editor” section, and submit your "Accept" recommendation.

Reviewer #3: All comments have been addressed

Reviewer #4: All comments have been addressed

2. Is the manuscript technically sound, and do the data support the conclusions?

Reviewer #3: Yes

Reviewer #4: No

3. Has the statistical analysis been performed appropriately and rigorously? 

Reviewer #3: Yes

Reviewer #4: No

4. Have the authors made all data underlying the findings in their manuscript fully available?

Reviewer #3: Yes

Reviewer #4: Yes

5. Is the manuscript presented in an intelligible fashion and written in standard English?

Reviewer #3: Yes

Reviewer #4: No

6. Review Comments to the Author

Reviewer #3: All comments have been addressed successfully by the authors.

The manuscript may be accepted for publication in the journal.

Reviewer #4: 1. As author said in the abstract that 156 approved cancer therapies based on small molecules are there so they should give reference for this.

2. Rationale behind the selection of CR80 is not satisfactory, Authors must give an explanation

3. The IC 50 values are in nanomolar and some in millimolar, authors should explain why they are different. Authors must use only one unit for all the compounds.

4. Authors should give numbering to the structures of scheme.

5. Authors should incorporate 3D representation of the docking results.

6. Authors must compare the binding affinity of the compounds with the standard drug.

7. The word in vivo, in vitro, in silico must be in italics.

8. The graphical representation is missing.

9. Authors must give graphs of IR, NMR and Mass of CR80.

10. Authors must follow the referencing style given by journal.

7. PLOS authors have the option to publish the peer review history of their article (what does this mean? ). If published, this will include your full peer review and any attached files.

**Do you want your identity to be public for this peer review?** For information about this choice, including consent withdrawal, please see our Privacy Policy .

Reviewer #3: **Yes**

Reviewer #4: No

---

## [Author Response · Author response to Decision Letter 2]

30 Jan 2025

Sapan Kamleshkumar Shah, Ph.D., M.Pharm.

Academic Editor

PLOS ONE

Additional Editor Comments and Reviewer 4

Thank you for your comments, which enable us to present an improved version of the manuscript. Below, we provide a point-by-point response to the comments from the editor and referee 4.

The manuscript is ready to accept and accepted after some minor corrections.

1. As the author said in the abstract, 156 approved cancer therapies based on small molecules are available, so they should give a reference for this.

Response: reference was added.

3. National Cancer Institute. Cancer Drugs - National Cancer Institute [Internet]. www.cancer.gov. 2012. Available from: https://www.cancer.gov/about-cancer/treatment/drugs.

2. The rationale behind the selection of CR80 is not satisfactory; authors must give an explanation

Response: Thank you for your comment; we would like to clarify why the selection of CR80

We want to clarify the reasons for selecting CR80. The candidate's selection was not solely based on one criterion. Our proposal suggests that, regarding Hit CBZ, the pharmacodynamic and pharmacokinetic attributes would see improvement. Candidates 57, 58, 73, and 79 have log values greater than 4, making them acceptable, whereas candidate 70 does not demonstrate sufficient affinity; therefore, we did not select the mentioned candidates in this instance.

Additionally, the criteria for choosing the designed molecules included the following: the IC50 values should be less than ten micromolar, the affinity for the molecular target must be more negative than -8 kcal/mol (considering that the CBZ nucleus has a value of -7 kcal/mol), and the logP values should be within the range of 2 to 4, as this range has been shown to yield promising experimental results. Lastly, the toxicity profile should be comparable to or better than that of existing drugs. Consequently, any candidates that had a more promising toxicity profile but did not satisfy the three primary criteria—affinity, IC50, and logP—were excluded from consideration.

3. The IC 50 values are in nanomolar and some in millimolar. The authors should explain why they are different. They must use only one unit for all the compounds.

Response: Thanks for the suggestion; the change of units referred to a way of representing a low concentration, but all the units and the graph have already been converted to micromolar.

4. Authors should give numbering to the structures of the scheme.

Response: this was corrected

5. The words in vivo, in vitro, and silico must be in italics.

Response: this was corrected in the manuscript.

6. Authors must provide graphs of IR, NMR, and Mass of CR80.

Response In the SI, IR, and MS-HR were added. NMR was already in the SI

7. Authors should incorporate a 3D representation of the docking results.

Scheme 2. Non-covalent interactions CR80 and 3D representation

8. Authors must compare the binding affinity of the compounds with the standard drug.

Response: The manuscript in the following paragraph is compared to the standard drug described.

“We used the Beta-tubulin structure with the code 6QUS as the molecular target, obtained from the RCSB Protein Data Bank (RCSB PDB). This protein is a microtubule-organizing protein that specifically binds to the minus end of non-centrosome microtubules and regulates their dynamics and organization. Vincristine was used as a validation ligand with demonstrated activity and compared with the designed analogs (entry 2 table 1)[31].”

9. The graphical representation is missing.

This information can be found in an additional file titled "Figures and Schemes."

10. Authors must follow the referencing style given by the journal.

The referencing style has been corrected.

James Guevara Pulido

Corresponding author

joguevara@unbosque.edu.co

---

## [Editor Report · Decision Letter 2]

2 Feb 2025

Design, Synthesis, and In Vitro Evaluation of a Carbamazepine Derivative with Antitumor Potential in a Model of Acute Lymphoblastic Leukemia

PONE-D-24-38959R2

Dear Dr. Guevara Pulido,

We’re pleased to inform you that your manuscript has been judged scientifically suitable for publication and will be formally accepted for publication once it meets all outstanding technical requirements.

Kind regards,

Sapan Kamleshkumar Shah, Ph.D., M.Pharm.

Academic Editor

PLOS ONE

Additional Editor Comments (optional):

Authors have implemented all suggestions by reviewers and recommend for further publication.
---

## [Editor Report · Acceptance letter]

PONE-D-24-38959R2

PLOS ONE

Dear Dr. Guevara Pulido,

I'm pleased to inform you that your manuscript has been deemed suitable for publication in PLOS ONE. Congratulations! Your manuscript is now being handed over to our production team.

Kind regards,

on behalf of

Dr. Sapan Kamleshkumar Shah

Academic Editor

PLOS ONE